# A Molecular Model of PEMFC Catalyst Layer: Simulation on Reactant Transport and Thermal Conduction

**DOI:** 10.3390/membranes11020148

**Published:** 2021-02-20

**Authors:** Wenkai Wang, Zhiguo Qu, Xueliang Wang, Jianfei Zhang

**Affiliations:** Moe Key Laboratory of Thermo-Fluid Science and Engineering, Energy and Power Engineering School, Xi’an Jiaotong University, Xi’an 710049, China; wangwenkai@stu.xjtu.edu.cn (W.W.); xlwang082@mail.xjtu.edu.cn (X.W.); zhangjf@mail.xjtu.edu.cn (J.Z.)

**Keywords:** catalyst layer, MD simulation, oxygen transport, thermal conductivity, PEMFCs

## Abstract

Minimizing platinum (Pt) loading while reserving high reaction efficiency in the catalyst layer (CL) has been confirmed as one of the key issues in improving the performance and application of proton exchange membrane fuel cells (PEMFCs). To enhance the reaction efficiency of Pt catalyst in CL, the interfacial interactions in the three-phase interface, i.e., carbon, Pt, and ionomer should be first clarified. In this study, a molecular model containing carbon, Pt, and ionomer compositions is built and the radial distribution functions (RDFs), diffusion coefficient, water cluster morphology, and thermal conductivity are investigated after the equilibrium molecular dynamics (MD) and nonequilibrium MD simulations. The results indicate that increasing water content improves water aggregation and cluster interconnection, both of which benefit the transport of oxygen and proton in the CL. The growing amount of ionomer promotes proton transport but generates additional resistance to oxygen. Both the increase of water and ionomer improve the thermal conductivity of the C. The above-mentioned findings are expected to help design catalyst layers with optimized Pt content and enhanced reaction efficiency, and further improve the performance of PEMFCs.

## 1. Introduction

Proton exchange membrane fuel cells (PEMFCs) have attracted attention from energy devices such as portable, mobile, and stationary devices because it helps effective reductions of energy shortage and environment pollution [1,2,3,4]. However, the high price of PEMFCs components, especially the platinum (Pt) group catalyst becomes one of the bottlenecks that limit their commercial development. Numerous strategies and approaches have been tried to reduce the catalyst platinum loading, which is still hard to meet the Department of Energy (DOE) target of a cathode Pt loading of 0.1 mg/cm^2^ [5]. In pursuit of reducing platinum loading and raising power efficiency, many scholars have found that the resistance of oxygen transmission is becoming particularly obvious under low platinum loading and high current density, which greatly limits the oxygen reduction reaction (ORR) and hence reduces the current density. At present, the studies about reactant transport and reaction efficiency in CL are in the primary stage. Gasteiger et al. [6] quantified the activation loss and voltage loss and proposed two theoretical methods to test the performance of catalysts. Some of the tested non-Pt catalysts have a 2.5–4 times enhancement of mass activity. Wonseok et al. [7] studied the influence of a layer of a thin film on the surface of catalysts on oxygen transport and battery performance using an agglomerate model. The review of Weber et al. [8] summarized the sharp resistance increase when Pt capacity was less than 0.1 mg/cm^2^ from the experimental and numerical model results. In general, the local resistance of oxygen is the sum of many parts when thronging the multi-components of the catalyst layer. However, the mathematical model and experimental measurement method for the quantitative and qualitative analyses have been not unified due to the different morphologies, structures, and complex micro-interaction [9,10,11,12].

As for numerical simulation research, analytical models [13,14], mesoscopic methods such as lattice Boltzmann method (LBM) [15,16], microscopic methods such as molecular dynamics (MD) [17,18,19,20,21,22,23,24,25,26,27] simulation, and density functional theory (DFT) [28,29,30,31] are usually used to construct the CL structure and study the mechanisms of reactant transport, reaction, and heat conduction. For example, Liang et al. [14] proposed the fractal theory of porous media to quantify the effective electrolyte diffusivity in porous media with consideration of the electrical double layer (EDL) effects. Chen et al. [15,16] proposed a “watermelon model” containing primary and secondary pores to simulate the transport under different conditions. Rao et al. [18,19] established the molecular dynamics model of polymer membrane to study the proton and thermal conductivity. Feng et al. investigated the thermal-mechanical properties in [20] and gave the thermal expand coefficients with different Pt–C combination styles. She also studied the multicomponent interaction and mechanical capacity in [21] and obtained the stress–strain characteristics with various water contents. Fan et al. [22] expounded that over 90% of O_2_ transportation occurred at the upper corner and edge regions rather than the faces of Pt because of the different arrangement tightness of perfluorosulfonic. He also studied the effects of side chain length on the thermal conductivity of the perfluorosulfonic acid (PFSA) membrane in another paper [26]. Zhao et al. [27] used the oxidized graphite to design a three-phase microenvironment in CL by MD simulation and experiments. Taeyoon et al. [29] calculated the adsorption characteristics and electron transfer of the catalysts with optimized geometry using the density functional theory (DFT) with carbon nanotube added in the CL. However, there are some deficiencies in the model establishment and parameter calculation based on the above research analysis. Firstly, some micro-structure are not built corresponding to the real components such as the carbon pores. Secondly, the parameter study is not so comprehensive from the molecular-level. Moreover, there is a paucity of research about the thermal conductivity in the CL using MD simulations. The MD simulation method make the parameter study enable profiting from the modifiable number of molecules and state of operation. For the catalyst layer, the parameter study contains multiple structural parameters and operating parameters. In this study, two typical parameters—water content and ionomer-to-carbon ratios—are used to investigate their effect on transport and thermal conduction in CL. In this study, a molecular model containing the multicomponent cathode catalyst layer is built and the corresponding equilibrium MD and nonequilibrium MD simulations with different water contents and ionomer-to-carbon ratios are conducted. The radial distribution functions (RDFs), mean square displacements (MSDs), and water contour distribution are calculated to illuminate the relationships between each component. The thermal conduction characteristics are obtained via the given heat flux and temperature gradients. This work is expected to help understand the multicomponent nanostructures, transport properties, and thermal features of the catalyst layer in PEMFCs.

## 2. Materials and Methods

### 2.1. Model and Dynamics Simulation Details

The cathode CL is composed of carbon-supported Pt particles, water, hydronium (proton), oxygen, and ionomer. Figure 1 shows the schematic construction of the PEMFC catalyst layer between the proton exchange membrane (PEM) and gas diffusion layer (GDL). In the CL the interactive multicomponent mixture gathers around the catalysts after oxygen transport through the GDL, and hydronium ions through the membrane. After a complex transport process at the three-phase interface of the reactants, the reaction takes place near the catalyst reactive place and produces water. In order to have a comprehensive reflection of the complex microstructures and the multi-components reactions in the catalyst layer, a molecular model of CL is built as shown in Figure 2. The diameter of carbon particles is 5 nm, and each of them contains 8000 carbon atoms. The pores inside the carbon particles, which are the so-called primary pores and the secondary pores, have been set to make the results closer to reality. A total of 4–6 Pt particles with 1.2 nm diameter are dispersed randomly on the carbon particle surface and inner pore. The mass fraction of Pt is 25 wt% in the Pt–C particle system. 

In addition to the Pt–C particles, the ionomer is an important part of CL owing to its proton conduction effects. Although some new short-chain polymers have been used in the catalyst layer such as adding aromatic groups. In addition, it is noted that some properties would change like ionic conductivity, oxygen resistance, and thermal conductivity with different polymers. This study focuses on the reactant transport and thermal conduction at the specific polymer. Hence, Nafion polymer, which is the most commonly used polymer in an experimental and commercial application, is adopted in this work. The chemical structure of the ionomer monomer from the Nafion polymer is demonstrated in Figure 3. In this study, *m* = 1, *n* = 2, *x* = 6, and *y* = 1, and each ionomer particle contains 10 monomers. To investigate the influence of water content and ionomer quantities on the reactants′ transport and thermal conductivity, the various water contents are set as 1, 4, 7, and 10, and ionomer-to-carbon ratios are 0.4, 0.8, 1.2, and 1.6. As for the determination of I/C ratios, the commercial catalyst ionomer content is about 10–30%. In addition, this content can reach 50% to the maximum in some reported experimental studies. After unit conversion, the I/C ratios of 0.4, 0.8, 1.2, and 1.6 used in this paper are 24%, 39%, 49%, and 56% ionomer content. The I/C = 0.4–1.2 are within the scope of the commercial and experimental application while preparing the catalyst layer. Moreover, in order to obtain the influence rules with a slightly excessive ionomer, the I/C= 1.6 are also set as the simulation cases. The detailed molecule numbers of each component in the simulations are listed in Table 1 and Table 2. 

All simulations in this work are conducted using Materials Studio 2017 molecular modeling software (Accelrys, San Diego, CA, USA), and the COMPASS force field is used, which can calculate uniformly the organic molecular system and the inorganic molecular system covering all the adopted molecules in the simulation. 

In the COMPASS force field, total energy between the atoms contain the valance energy and the non-bond energy [32], which can be expressed as follows:(1)$$E_{total}=E_{valence}+E_{non-bond}$$

The valance energy contained the bond item, angle item, torsion item, and out-of-plane vibration item in Equation (2). The non-bond energy contained the Van der Waals item and the Columbia item in Equation (3).
(2)$$\begin{array}{l} E_{valence}=\sum_{b}\left[k_{2}{\left(b-b_{0}\right)}^{2}+k_{3}{\left(b-b_{0}\right)}^{3}+k_{4}{\left(b-b_{0}\right)}^{4}\right]+\sum_{\theta }\left[k_{2}{\left(\theta -\theta_{0}\right)}^{2}+k_{3}{\left(\theta -\theta_{0}\right)}^{3}+k_{4}{\left(\theta -\theta_{0}\right)}^{4}\right]\\ +\sum_{\varphi }\left[k_{1}\left[1-\cos(\phi -\phi_{0})\right]+k_{2}\left[1-\cos(2\phi -\phi_{0})\right]+k_{3}\left[1-\cos(3\phi -\phi_{0})\right]\right]+\sum_{\chi }k_{2}\chi_{2}\\ +\sum_{b,b\prime }k(b-b_{0})(b\prime -b_{0}\prime )+\sum_{b,\theta }k(b-b_{0})(\theta -\theta_{0})+\sum_{b,\phi }\left(b-b_{0}\right)\left[k_{1}\cos\phi +k_{2}\cos2\phi +k_{3}\cos3\phi \right]\\ +\sum_{\theta ,\phi }\left(\theta -\theta_{0}\right)\left[k_{1}\cos\phi +k_{2}\cos2\phi +k_{3}\cos3\phi \right]+\sum_{b,\theta }k(\theta -\theta_{0}\prime )(\theta -\theta_{0})+\sum_{\theta ,\theta ,\phi }k(\theta -\theta_{0}\prime )(\theta -\theta_{0})\cos\phi \end{array}$$
(3)$$E_{non-bond}=E_{vdW}+E_{coulombic}$$
(4)$$E_{vdW}=\sum_{i,j}\varepsilon_{ij}\left[2{\left(\frac{r_{ij}^{0}}{r_{ij}}\right)}^{9}-3{\left(\frac{r_{ij}^{0}}{r_{ij}}\right)}^{6}\right]$$
(5)$$E_{coulombic}=\sum_{i>j}\frac{q_{i}q_{j}}{r_{ij}}$$
where *b*, *θ*, and *Φ* represent the bond length, bond angle, and dihedral angle, respectively, and *χ* is the heterogeneous shape parameter.

### 2.2. Analysis Theory

#### 2.2.1. Radial Distribution Function 

To find the interaction between the multicomponent mixture in the CL, the radial distribution function (RDF) is calculated by Equation (6) as follows:(6)$$g_{A-B}(r)=\frac{Vn_{B}}{4N_{B}\pi r^{2}dr}$$
where *V* represents the volume of the simulation system volume and *n_B_* is the number of atom B located in a spherical shell centered on atom A. The shell’s volume is *4πr^2^dr. N_B_* is the number of B atoms in the system. 

#### 2.2.2. Mean Square Displacement and Diffusion Coefficient

The mean square displacement (MSD) can be expressed as Equation (7).
(7)$$MSD(t)=\frac{1}{N}\left({\sum_{i=1}^{N}\left|r_{i}(t)-r_{i}(0)\right|}^{2}\right)$$

In addition, the diffusion coefficient *D* could be obtained by the Einstein formula as follows: (8)$$D=\frac{1}{6N}\underset{t\to \infty }{\lim}\frac{d}{dt}\sum_{i\to j}N{\left|r_{i}(t)-r_{i}(0)\right|}^{2}$$

#### 2.2.3. Thermal Conductivity

The thermal conductivity is calculated using the non-equilibrium molecular dynamics (NEMD) [33,34] method in which the energy flux is generated by exchanging the kinetic energy of two particles (in this paper are atoms). After a certain number of times of momentum exchanges, the average heat flux is obtained via the total energy exchange divided by response time and cross-sectional area. As shown in Figure 4, the heat flux is imported Input inward the system from both ends of the model and the temperature gradient is provided via a statistic of the interior temperature after the system reaches stability and then the thermal conductivity is calculated by the Fourier law of heat conduction as follows:(9)$$K=-\frac{J_{Z}}{\left(\frac{dT}{dz}\right)}$$
where *dT*/*dz* is the temperature gradient. Because the direction of the flux is opposite from the gradient, the thermal conductivity is always positive. *J_Z_* represents the energy flux in the Z-direction which can be expressed as
(10)$$\left|J_{Z}\right|=\frac{\Delta E}{2A\Delta t}$$
where ∆*t* is the interval time of every kinetic energy exchange and ∆*E* is the generating energy between two layers. *A* is the area of XY-direction of the model. Factor *2* is due to periodic boundary conditions so the amount of heat flux flows at either side of the model is *E/2*.

The three-dimensional period model is 200 Å in the Z direction and 50 × 50 Å in the XY direction (the aspect ratio is 4:1), as recommended in the references. In addition, the model is divided into 40 layers, as shown in Figure 4. The time step is 0.5 fs. First, the geometry optimization of the system is conducted to obtain energy minimization, followed by an equilibration stage with a thermostat for 200 ps. Finally, MD simulations are conducted for 500 ps at 300 K in the NVT ensemble, and the trajectory and dynamical properties are recorded for the thermodynamic analyses.

## 3. Results and Discussion

### 3.1. Effects of Water Content and Ionomer-to-Carbon Ratio on RDFs

Figure 5 depicts the variation of RDF as a function of water content in the catalyst layer. Specifically, Figure 5a shows flat and wide peaks in all these Pt–C curves, indicating the uniformity of the distribution of Pt in the supported carbon. The minimum concatenate distance between carbon and Pt is set at 2.8 Å in both the inner and outer pores. It is clear that the RDF value first increases and then decreases with the λ ranges from 1 to 10, indicating that the increasing water facilitates the adsorption of Pt in carbon but the excess water reverses this trend. The RDF values shown in Figure 5b indicate that the Pt–H^+^ distribution is not influenced by the water contents. Unlike the H^+^, Figure 5c shows that the g_Pt-O_(r) increases with increasing λ, which illustrates that water promotes the accumulation of O_2_ near the catalyst. The RDF of sulfonate–sulfonate represents the distribution homogeneity of the sulfonic acid group, and it can be seen from Figure 5d that with the increase of λ, the short-range effect of sulfonate decreases, indicating that the ionomer uniformity is enhanced. The agglomeration phenomenon is obvious when the water content is very small (λ = 1). The RDF values of Pt–S in Figure 5e show that the water content has little effect on the ionomer distribution around the Pt–C surface. In Figure 5f,g, both of the RDF values of S–H_2_O and S–H_3_O^+^ decrease with increasing water content, and the absolute value of g_S-H3O+_(r) is higher than that of the S–H_2_O. This indicates that water promotes the aggregation of H^+^ around the acid group, and the degree of aggregation is much higher than that of water, which can be attributed to the ion adsorption and desorption effect of cation and anionic. It is worth noting that the second peaks of S–H_3_O^+^ RDFs are the hydrogen bond interaction between the sulfur atom and hydronium ion. Therefore, when λ = 1, the peak is missing because of the lack of hydronium ion. The g_O-O_(r) in Figure 5h is small when λ = 1 and becomes higher when the λ continues to increase, indicating that the water content has little effect on the oxygen short-range order except for the state of extreme lack of water. To summarize, the increase of water is beneficial to the Pt–C combination and the accumulation of O_2_ near the catalyst. It also reduces the aggregation effect of water and hydronium ion around the sulfonic acid ion. The promotion effects are in accordance with Feng et al simulation results [21] and further lift the current density as experimental data showed in Carcadea et al. [35]. 

Figure 6 depicts the variation of RDF as a function of the ionomer-to-carbon ratios in the catalyst layer. Figure 6a indicates that the RDF values of Pt–C slightly increase as the I/C increase and then remain constant when further increases the ionomer, which confirms that an optimized ionomer content can benefit the combination of Pt and carbon. Figure 6b shows the RDF values of Pt–H^+^ are not affected by the I/C, which is similar to the effect of water content. Figure 6c shows that the g_Pt-O_(r) decreases with an increasing ionomer, which illustrates that the ionomer impedes the accumulation of O_2_ near the catalyst and will further be averse to the reaction. Figure 6d shows that the S–S short-range order becomes higher with the high ionomer content. However, the peak values are not multiplied in accordance with the increasing sulfonate group. Figure 6e shows a slight decline of the g_Pt-S_(r), illustrating that the increase of ionomer will not cluster around the catalysts and therefore it is not advantageous to the proton transport. Figure 6f shows that g_S-H2O_(r) values are almost the same. The different influences in the sulfonate–water distribution induced by water content and ionomer content can be explained in terms of that the water is adequate. As can be seen in Figure 6g, the values of g_S-H+_(r) first increase with the I/C ranging from 0.4 to 1.2 and then decline when the I/C reaches 1.6. This can be attributed to the relative absence of enough hydronium ion. The results of O–O values indicate that little effect on the oxygen short-term order is induced by varying the value of I/C, which is similar to the effect of water content. To summarize, the increase of ionomer can a little bit promote the Pt–C combination. However, it impedes the transportation of oxygen to Pt and proton around the sulfonic group. The rules fit well with Carcadea and Huang et al. experimental results [35,36], in which the current density did not significantly increase when ionomer increase.

### 3.2. Effects of Water Content and Ionomer-to-Carbon Ratio on Diffusion Coefficients

According to Equations (6) and (7), the diffusion coefficients are calculated of various water contents and the results are shown in Figure 7. To have a better understanding of the properties and mechanism of the oxygen and hydronium ion diffusion, the water cluster morphology in the catalyst layer is worth studying. In this paper, the water cluster morphology is given via the water molecular density contours after the dynamics simulation. Moreover, the connecting threshold is set to 2 Å, referring to the 3.5 Å in literature [27] and 1.4 Å in [37]. The water cluster morphology diagrams are demonstrated in Figure 8 and the yellow isosurface represents the water clusters region. In Figure 7, the diffusion coefficient of oxygen first reduces from 2.6 × 10^−6^ cm^2^/s to 2.1 × 10^−6^ cm^2^/s and then increases to 3.5 × 10^−6^ cm^2^/s with increasing water contents. Moreover, the diffusion coefficient of the proton shows an increase of 0.1, 0.25, 0.3, and 0.36 (10^−6^ cm^2^/s) with the corresponding λ of 1, 4, 7, and 10, respectively. The D_O2_ is an order of magnitude larger than the D_H+_, which is in good agreement with the D_O2_ in literature [21] and D_H+_ in [38]. Due to the measurement difficulties in experiments and modeling error in simulation methods, the same order of magnitude of error level can be acceptable. Figure 8 shows the variations of water clusters form in the CL system. When the λ = 1, the hydronium ions are present in discontinuous spots. According to the RDF results of S–H_3_O^+^ mentioned in Section 3.1, these water clusters tend to spread around the sulfonic acid group. With increasing water content, the clusters volume become grew and interconnected until all the water formed continuous pathways (λ = 10). As the S–H_2_O RDF values display in Figure 5f, the water will spread evenly in the CL system rather than merely around the ionomer when λ = 10. The reason why the D_H+_ is an order of magnitude smaller than oxygen can be explained as the received stronger interaction from the sulfonate anion. The addition of D_H+_ is due to that the excessive water declines this kind of interaction as verified in Figure 5f.

The variation of the D_O2_ can be explained in terms of the three diffusion ways of oxygen in the CL system—diffusing in water, diffusing in gas, and diffusing across the gas-water interface. In fact, the “gas” refers to the space that is not occupied by water and ionomer. When the λ is quite small, the diffusion of O_2_ mainly occurs in the gas region so the diffusion coefficient is relatively large. As the water increase, the area of the gas–water interface increases, which further results in a higher oxygen diffusion resistance, as depicted in Figure 8b,c. Therefore, the D_O2_ reduces when the λ= 4 and λ = 7. With a continuous increase in the water content (λ = 10), the connected water pathways increases and the area of the gas–water interface becomes smaller, resulting in a remarkable increase in the D_O2_. In summary, the oxygen diffusion is affected to a great extent by the water cluster states and their connected area. The proton diffusion is only affected by the water content.

The effects of the ionomer-to-carbon ratio on the reactant diffusion coefficients and water cluster morphology are depicted in Figure 9 and Figure 10, respectively. With increasing I/C ratio, the D_O2_ value declines from 2.2 to 2.1 (10^−6^ cm^2^/s), 2.05, and 1.7 (10^−6^ cm^2^/s), while the D_H+_ increases with from 0.05 to 0.08, 0.3, and 0.36 (10^−6^ cm^2^/s), respectively. The results imply that the addition of ionomer promotes proton diffusion but impedes the transportation of oxygen. The promotion to proton transport is understandable because of the sulfonic acid conduction carrier. The variation of D_O2_ can also be explained by the water morphology in Figure 10. It can be seen in Figure 10 that with increasing ionomer content, the connected pathways become less, and then the independent clusters occur. According to the S–H_2_O RDF depicted in Figure 6f, these clusters will be center on the sulfonic acid group all the time. It should be noted that the declining extent of D_O2_ is higher than water, implying a more obvious oxygen impediment effects from ionomer. On the contrary, the promotion effect of ionomers on the proton conductivity is more significant than that of water. In summary, the effects of ionomer increase on oxygen transport will also be influenced by the water formation in CL.

### 3.3. Effects of Water Content and Ionomer-to-Carbon Ratio on Thermal Conductivity

The atom momentum exchange NEMD program described in Section 2.2.3 is conducted to calculate the energy flux and temperature gradient, which is further utilized to obtain the thermal conductivity coefficients. Figure 11a shows the thermal conductivity of the catalyst layer as a function of water content. With increasing water content, the thermal conductivity coefficients increase gradually from 0.5 to 0.8, 1.06, and 2.5 (W/(m·K)). It should be noted that the result in this study accords with the studies of Kang et al. [39] and Jiang et al. [40]. This increment can be attributed to a huge variation of the thermal conductivity of each component in CL, and the Pt–C system has a higher thermal conductivity value than others. In the periodic volume-fixed computational domain, the volume fraction of water enlarges, while the volume fractions of the other components decrease with increasing λ. The inconsistent amount of these decreases makes the volume fraction of the Pt–C bigger; hence, the thermal conductivity goes up. The result proves that the increasing water is beneficial to the enhancement of the heat conduction of the catalyst layer. The thermal conductivity variations under different I/C ratios are shown in Figure 11b. Similarly, the thermal conductivity coefficient increases gradually from 0.3 to 0.7, 1.06, and 1.6 (W/(m·K)). The effect of I/C ratios on the thermal conductivity is inapparent compared with λ variation, indicating that the promotion effect of ionomer is a slightly weaker than water.

Figure 12 shows the variation of temperature as a function of water content. It is found that the temperature gradient lines are not straight. The bending instances occur at the location where the Pt–C particles are dispersed. Because of the huge difference in thermal conductivity for Pt–C particles, the temperature gradient displays a sudden rise and fall in the whole system with increasing layers. One notable aspect is the effect of particle dispersion on the error bar value. Specifically, the error value becomes higher when the particles move closer to both the hot and cold ends of the model. Moreover, the error bars obtained in Figure 11 are higher than in studies by Zheng et al. [18] and Fan et al. [26], which can be attributed to the existence of carbon-supported Pt in this system model.

## 4. Conclusions

In this study, a molecular model containing Pt, carbon, ionomer, water, oxygen, and hydronium ion is built to investigate the effects of water content and ionomer-to-carbon ratio on the RDFs between components. The reactants′ diffusion properties and the thermal conductivity are analyzed after equilibrium MD and nonequilibrium MD simulations. The main conclusions are drawn as follows.

The increase of water is beneficial to the Pt–C combination and the oxygen accumulation near the catalyst. It also reduces the aggregation effect of water and hydronium ion around the sulfonic acid ion. The oxygen diffusion coefficient firstly grows and then declines, which is dependent closely on the water cluster morphology and connection area. In addition, the proton diffusion coefficient increases with increasing water content. The increase of ionomer impedes the transportation of oxygen to Pt and protons around the sulfonic group. The oxygen transport is retarded while the proton diffusion is promoted, which is also closely related to the water cluster morphology and ionic interaction. The thermal conductivity of CL can be enhanced by increasing the water and ionomer contents. On the basis of this conclusion, we suggest that the I/C ratio should be within reasonable bounds, and the best ratio is approximately 0.8 when designing the structure of the catalyst.

## Figures and Tables

**Figure 1 membranes-11-00148-f001:**
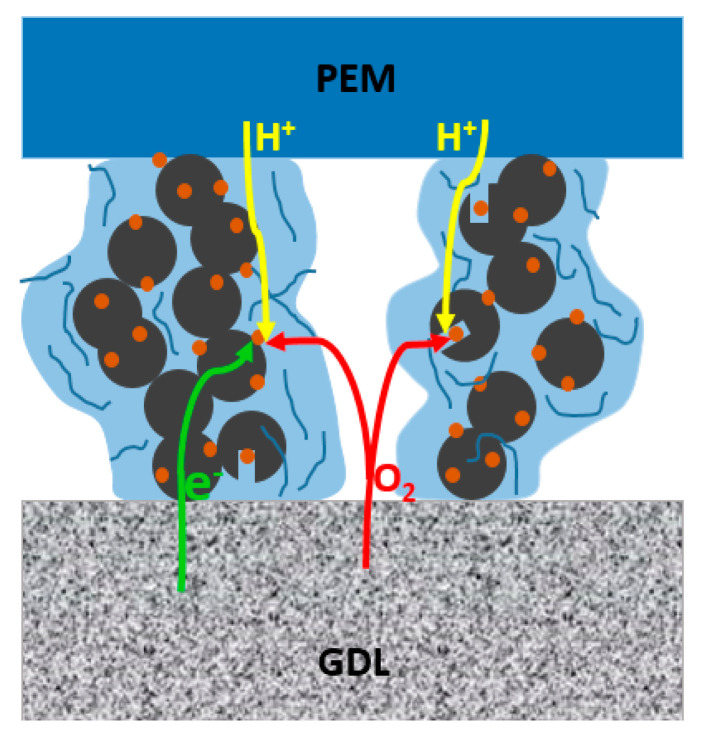
Schematic diagram of the proton exchange membrane fuel cell (PEMFC) catalyst layer between the PEM and GDL.

**Figure 2 membranes-11-00148-f002:**
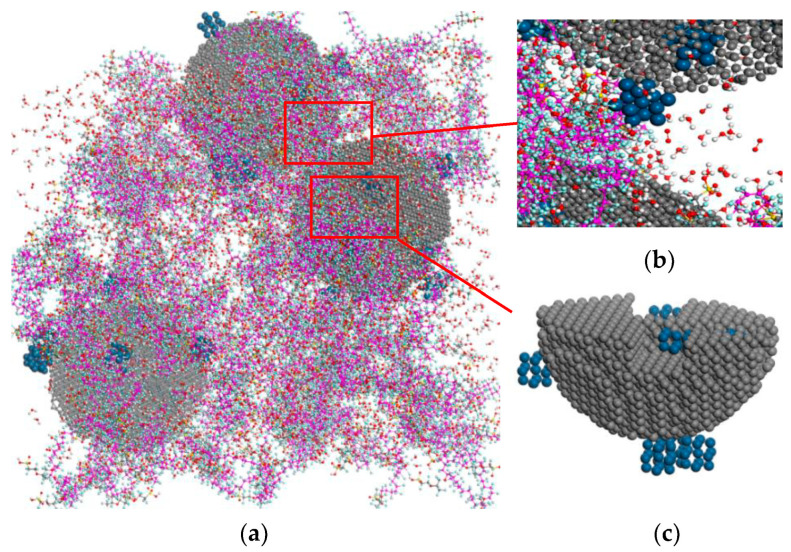
Molecular model of the multi-component catalyst layer between the PEM and GDL (**a**) molecular model of the catalyst later; (**b**) secondary pore; and (**c**) primary pore.

**Figure 3 membranes-11-00148-f003:**
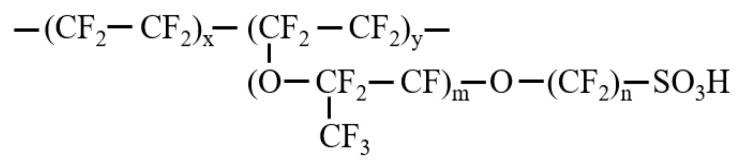
Chemical structure of the ionomer monomer.

**Figure 4 membranes-11-00148-f004:**
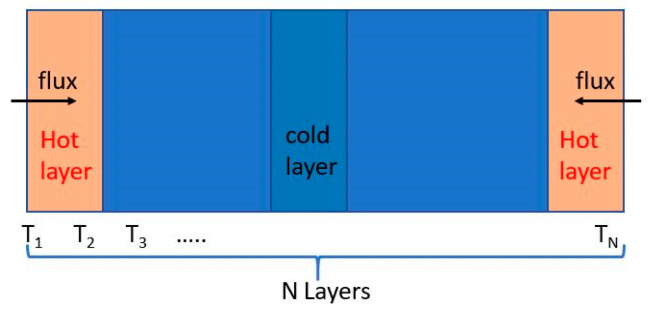
Schematic diagram of the heat transfer settings.

**Figure 5 membranes-11-00148-f005:**
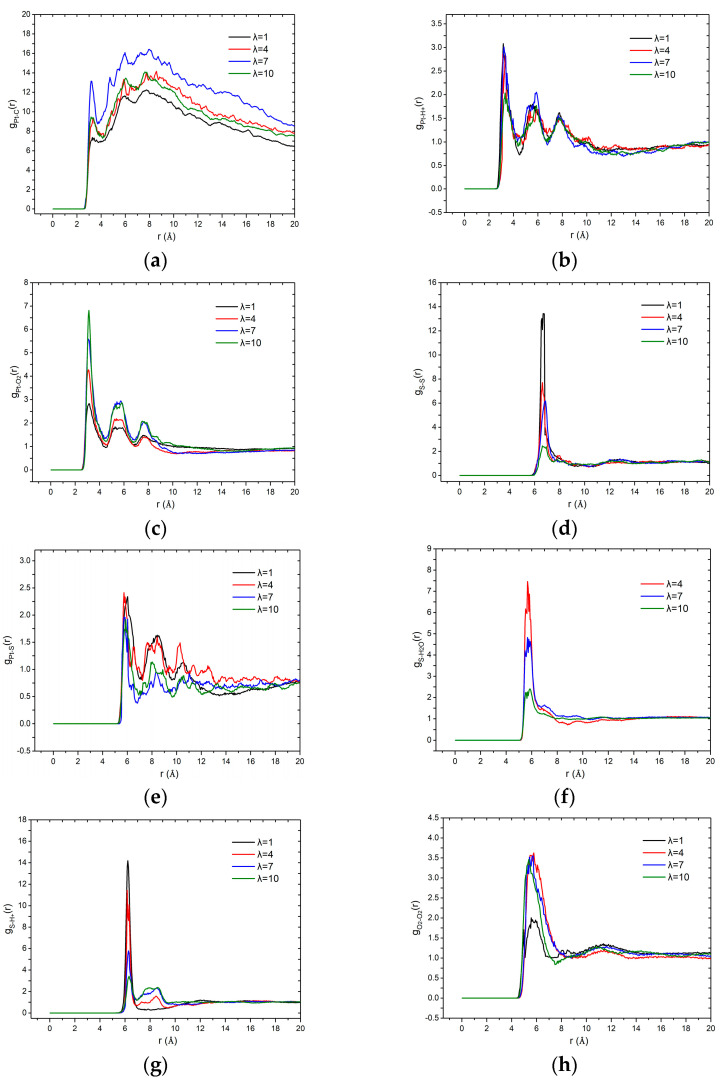
Radial distribution functions (RDF) values of the multi-components as a function of water content: (**a**) Pt–C; (**b**) Pt–H^+^; (**c**) Pt–O_2_; (**d**) S–S; (**e**) Pt–S; (**f**) S–H_2_O; (**g**) S–H^+^; and (**h**) O_2_–O_2_.

**Figure 6 membranes-11-00148-f006:**
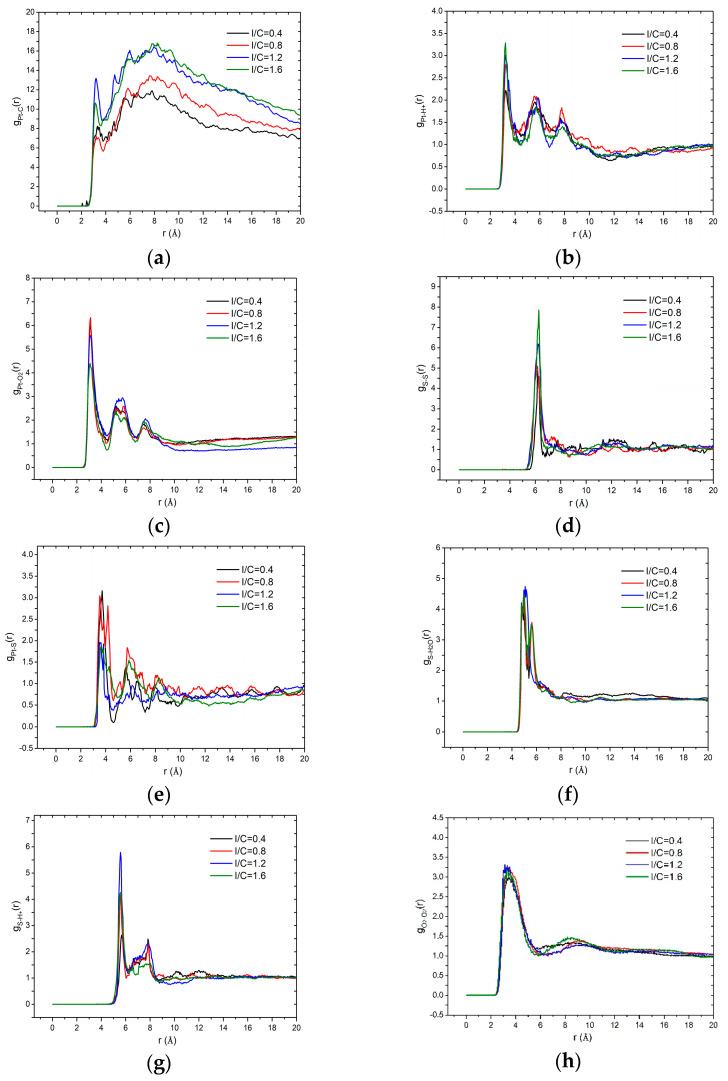
RDF values of the multi-components as a function of ionomer-to-carbon ratios: (**a**) Pt–C; (**b**) Pt–H^+^; (**c**) Pt–O_2_; (**d**) S–S; (**e**) Pt–S; (**f**) S–H_2_O; (**g**) S–H^+^; and (**h**) O_2_–O_2_.

**Figure 7 membranes-11-00148-f007:**
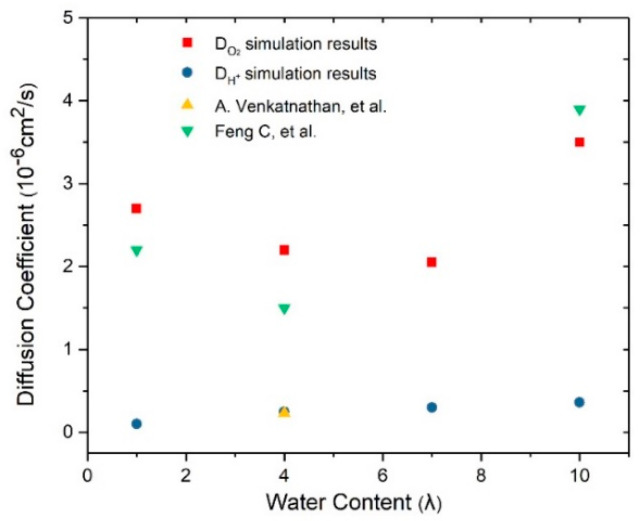
Diffusion coefficient of oxygen and hydronium ions as a function of water content.

**Figure 8 membranes-11-00148-f008:**
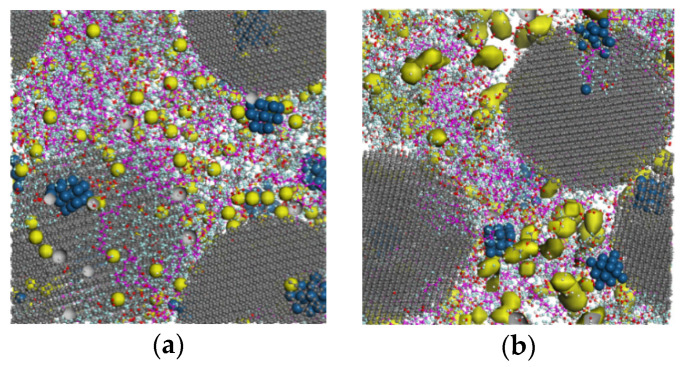
Water distribution and cluster form in catalyst layer (CL) at different water contents: (**a**) water clusters distribution at λ = 1; (**b**) water clusters distribution at λ = 4; (**c**) water clusters distribution at λ = 7; and (**d**) water clusters distribution at λ = 10.

**Figure 9 membranes-11-00148-f009:**
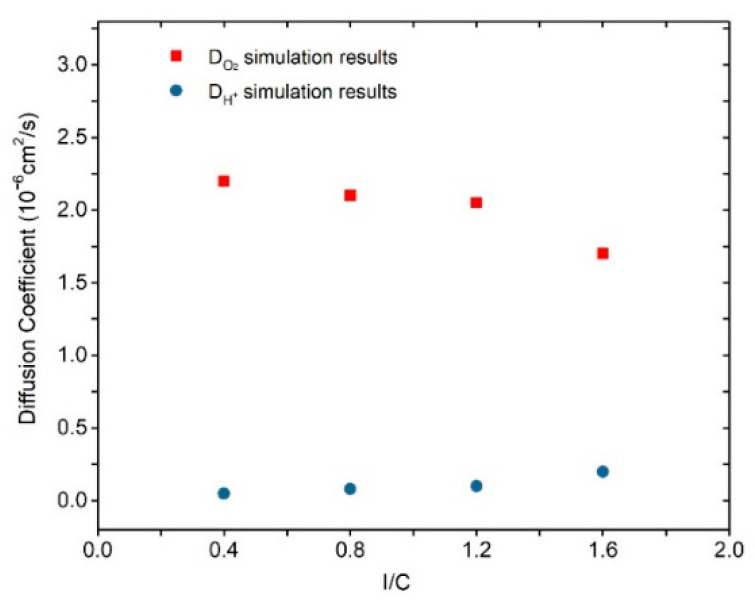
Diffusion coefficient of oxygen and hydronium ions at different ionomer-to-carbon ratios.

**Figure 10 membranes-11-00148-f010:**
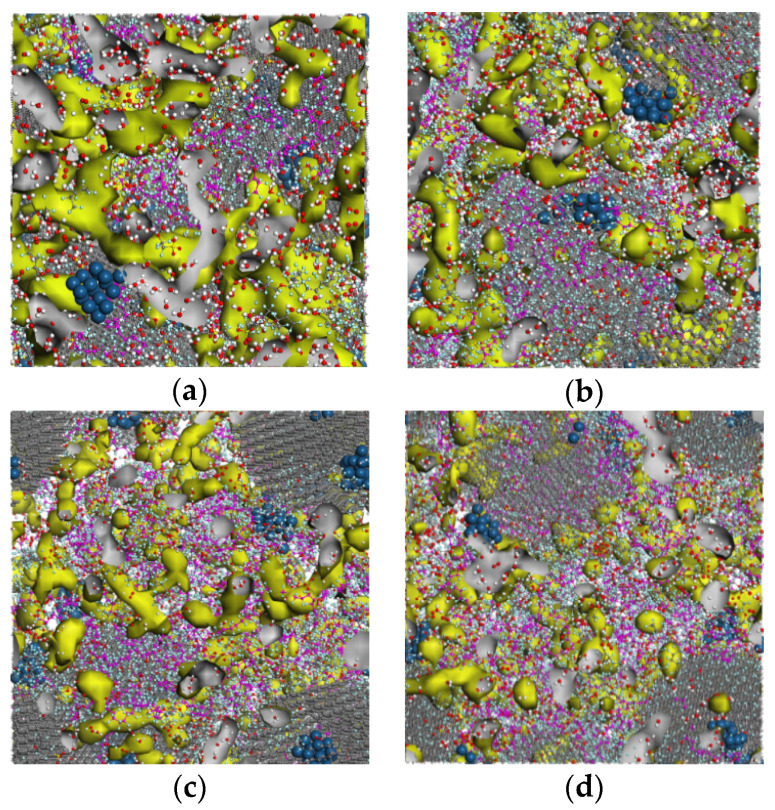
Water distribution and cluster form in CL at different ionomer-to-carbon ratios: (**a**) I/C = 0.4; (**b**) I/C = 0.8; (**c**) I/C = 1.2; and (**d**) I/C = 1.6.

**Figure 11 membranes-11-00148-f011:**
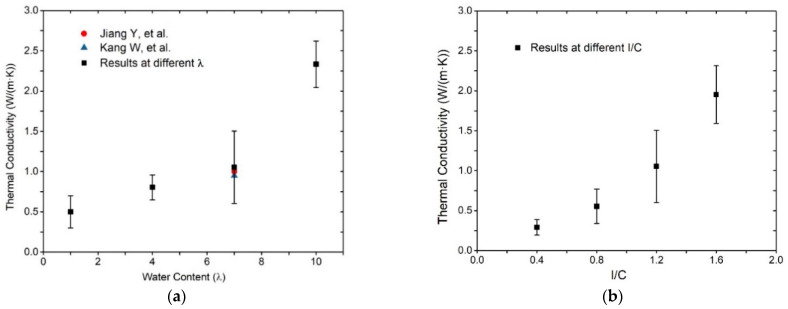
Thermal conductivity of CL with (**a**) different water contents and (**b**) different ionomer-to-carbon ratios.

**Figure 12 membranes-11-00148-f012:**
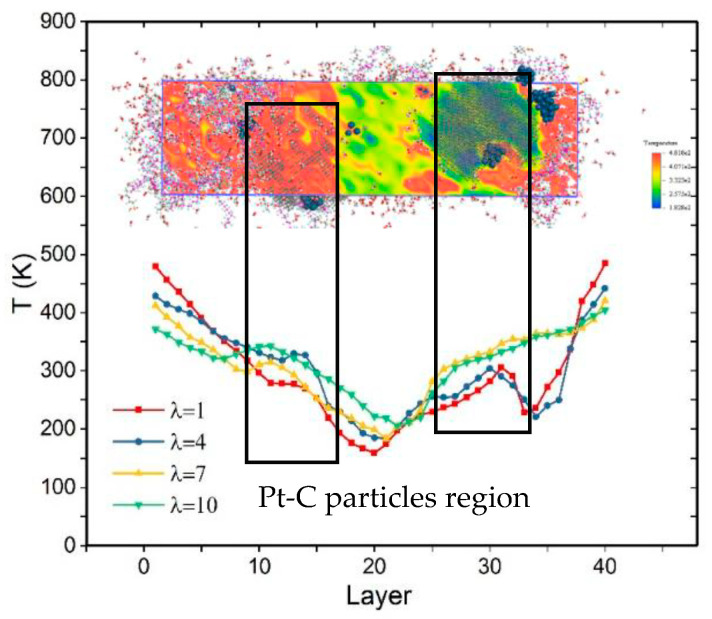
Temperature distribution of CL and its uneven gradient with different λ.

**Table 1 membranes-11-00148-t001:** Molecule numbers in models with different water contents.

	λ = 1	λ = 4	λ = 7	λ = 10
N_Ionomer_	30	30	30	30
N_Pt-C_	3	3	3	3
N_O2_	200	200	200	200
N_H2O_	0	600	1200	1800
N_H3O+_	200	200	200	200

**Table 2 membranes-11-00148-t002:** Molecule numbers in models with different ionomer-to-carbon ratios.

	I/C = 0.4	I/C = 0.8	I/C = 1.2	I/C = 1.6
N_Ionomer_	10	20	30	40
N_Pt-C_	3	3	3	3
N_O2_	200	200	200	200
N_H2O_	1200	1200	1200	1200
N_H3O+_	200	200	200	200

## Data Availability

Data available in a publicly accessible repository.

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
