# Peer review of "A Molecular Model of PEMFC Catalyst Layer: Simulation on Reactant Transport and Thermal Conduction"

_membranes, 2021, doi:10.3390/membranes11020148_

Round 1

Reviewer 1 Report

The research work reported in this manuscript addresses the effects of water content and ionomer-to-carbon ratio on the RDFs between components by building the molecular model containing Pt, carbon, ionomer, water, oxygen, and the hydronium ion. The authors also investigated the diffusion coefficient, water cluster morphology, and the thermal conductivity after the equilibrium MD and nonequilibrium MD simulations. This work could provide the industry some valuable information to optimize the catalyst layers design to improve the PEMFC performance. This manuscript can be considered for publication after minor revisions, considering the following comments/questions:

  1. In the paper, the authors used Nafion polymer as an ionomer in the catalyst layer. Currently, many new advanced polymer materials already employed in the catalyst layer, such as short-side-chain PFSA polymer with high IEC. In the molecular model presented in this paper, is it possible to also consider the effects of the types of the ionomer in CL since the ionomer with different IEC, water content and chemical structure/morphology could impact proton conductivity, water absorption et al.
  2. For Figure 3, the chemical structure of Nafion is not correct. Please double check.
  3. The various ionomer- to-carbon ratios are set as 0.4, 0.8, 1.2, 1.6 in your models. Why did you design the ratio range of 0.4-1.6? Typically, in a commercial catalyst coated membrane, the ionomer content in the catalyst layer is about 10-30%.
  4. The quality of Figures 5 and 6 need to be improved.
  5. 170-172, “To summarize, the increase of water is beneficial to the Pt-C combination and the accumulation of O2 near the catalyst. It also reduces the aggregation effect of water and hydronium ion around the sulfonic acid ion.” Suggested to cite some reference papers with the experimental data to support your conclusion.
  6. 176-178, the RDF values of Pt-C slightly increase as the I/C increase and then remain constant when further increases the ionomer, which confirms that an optimized ionomer content can benefit the combination of Pt and carbon. Suggested to cite some reference paper with the experimental data to support your conclusion.
  7. 264-265, “The effect of I/C ratios …… is a little bit weaker than water.” Please reword this sentence.
  8. What are your suggestions/comments to the PEMFC industry for catalyst layer design?

Reviewer 2 Report

The proton exchange membrane fuel cells (PEMFCs) are promising energy devices for stationary and mobile applications because of high power density, high efficiency, low operating temperature, low emissions, low noise, and great environmental compatibility. The PEMFCs are composed of gas diffusion layer (GDL) including gas diffusion backing (GDB) and microporous layer (MPL), membrane electrode assembly (MEA), and bipolar plates with gas channels. The fibrous gas diffusion layer is a core component of a PEMFC, enabling transport of gases, liquids and electricity within the cell. In this paper, a molecular model containing carbon, Pt, and ionomer compositions is built and the radial distribution functions (RDFs), diffusion coefficient, water cluster morphology, and thermal conductivity are studied after the equilibrium MD and nonequilibrium MD simulations. The topic may be important, the results are interesting and the methodology followed is appropriate, while the content falls well within the scope of this Journal. In general the paper makes fair impression and my recommendation is that it merits publication in this Journal, after the following major revision:

  1. The current one is nothing but a literature review. Why their work is important comparing to previous reports? I think this is essential to keep the interest of the reader.
  2. In Fig.7 and 9, the authors should give the explanations for the difference of data collected from different sources.
  3. Please check all Equations double.
  4. I am quite interested in some parametric study with the proposed molecular model. The manuscript could be more substantial if the authors do so. At least, the authors need to write some statements that how the proposed molecular model can be used for the parametric study.
  5. Proton exchange membrane fuel cells have attracted attention from energy devices such as portable, mobile and stationary devices, since it helps effective reductions of energy shortage and environment pollution. Besides molecular model, analytical models are very important tools, which can investigate proton exchange membrane fuel cells, see [International Journal of Hydrogen Energy, 2018, 43(37):17880-17888; International Journal of Heat and Mass Transfer, 2019, 137:365-371].
  6. At last, the English writing needs improving.

Reviewer 3 Report

Reviewer report on Manuscript Draft entitled ‘A molecular model of PEMFC catalyst layer:

 simulation on reactant transport and thermal conduction’

In this research authors have built a molecular model containing Pt, carbon, ionomer, water, oxygen, and hydronium ion. The model was applied to investigate the effects of water content and ionomer-to-carbon ratio on the radial distribution functions between components. The reactants' diffusion properties as well as the thermal conductivity were analysed after the simulations of equilibrium molecular dynamics and nonequilibrium molecular dynamics.

This investigation is interesting, from technological points of view. The research is in scope of the journal. Therefore, the manuscript eventually can be published after some minor corrections and improvements:

In introduction other types of nanocomposite-based membranes based on other polymers other metal nanoparticles and other types of catalysts (Formation and electrochemical evaluation of polyaniline and polypyrrole nanocomposites based on glucose oxidase and gold-nanostructures. Polymers 2020, 12(12), 3026) could be overviewed and discussed. Other possible application areas of here reported proton exchange membranes could be addressed and discussed in order to attract attention of readers from other research areas.

Discussion part of manuscript could be extended and advanced.

Round 2

Reviewer 2 Report

The Reference is very confusion. The citation of the text does not agree with Reference. For example, Ref.13 should be corrected as “Liang, M.; Liu, L.; Xiao, B.; Yang S.; Wang, Z.; Han, H. An analytical model for the transverse permeability of gas diffusion layer with electrical double layer effects in proton exchange membrane fuel cells. International Journal of Hydrogen Energy 2018, 43, 17880-17888, doi:10.1016/j.ijhydene.2018.07.186. ”

Ref.14 should be corrected as“Liang, M.; Fu, K.; Xiao, B.; Luo, L.; Wang, Z. A fractal study for the effective electrolyte diffusion through charged porous media. International Journal of Heat and Mass Transfer 2019, 137, 365-371, doi: 10.1016/j.ijheatmasstransfer.2019.03.141.”
